# Taste alteration and its relationship with nutritional status among cancer patients receiving chemotherapy, cross-sectional study

**Fatima Masoud Al-Amouri**[1], **Manal Badrasawi**[2]*

**1** Department of Nutrition and Food Technology, An-Najah National University, Nablus, Palestine,
**2** Department of Nutrition and Food Technology, Faculty of Agriculture and Veterinary Medicine, An-Najah National University, Nablus, Palestine

* m.badrasawi@najah.edu

**Data Availability Statement:** All relevant data are available on Figshare: Figures data: figshare, DOI: 10.6084/m9.figshare.25037483 Tables raw data: figshare, DOI: 10.6084/m9.figshare.25035959.

## Abstract

The aim of this study is to determine the prevalence of taste alterations (TAs) during chemotherapy and their association with nutritional status and malnutrition. In addition to the associated factors with TA, including sociodemographic health-related factors and clinical status, and to investigate coping strategies to manage TA. A multicenter cross-sectional design study was conducted on 120 cancer patients aged at least 18 who had been undergoing at least one round of chemotherapy. TAs were evaluated using the chemotherapy-induced taste alteration scale (CiTAS), the malnutrition universal screening tool (MUST) was used for nutritional screening, the antineoplastic side effects scale (ASES) was used for subjective assessment of chemotherapy side effects, and the Charlson comorbidity index (CCI) was used for comorbidity assessment. SPSS21 software was used to analyze the data, and the independent T-test and one-way ANOVA test were used to determine the association between TAs and a variety of related variables. The prevalence of TAs was 98.3%. Among participants, 48.3% were at low risk of malnutrition, 20% at medium risk, and 31.7% at high risk. Malnutrition risk was associated with taste disorders (p<0.05). Patients' age, gender, educational level, and physical status were associated with TAs (p<0.05). Type of cancer, chemotherapy regimen, and number of chemotherapy cycles were also associated with TAs (p<0.05). A variety of antineoplastic side effects were associated with TAs (p<0.05), including nausea, vomiting, dry mouth, sore mouth and throat, excessive thirst, swallowing difficulty, appetite changes, weight loss, dizziness, lack of energy, disturbed sleep, anxiety, and difficulty concentrating. TAs were associated with an increased number of comorbidities, and individuals with diabetes, pulmonary diseases, and hypertension were associated with TAs (P<0.05). Patients in this study rarely practice self-management strategies to cope with TAs. A high prevalence (98.3%) of TAs in cancer patients receiving chemotherapy was found, and it was linked to a variety of negative outcomes. Chemotherapy-induced TAs are an underestimated side effect that requires more attention from patients and health care providers.

**Funding:** The author(s) received no specific funding for this work.

**Competing interests:** The authors have declared that no competing interests exist.

## Introduction

Taste alteration is a negligible side effect in cancer patients. Oncology specialists underestimate it [1], and cancer patients underreport it despite its severity, prevalence, and implications. Patients rarely communicate it to their healthcare providers due to a lack of knowledge and difficulty recognizing and even describing the feelings they experience in the way they perceive tastes [2].

It is estimated that more than 75% of patients receiving chemotherapy reported that their food was too sweet, sour, salty, bitter, or tasteless, or even tasted like cardboard, metal, or sandpaper [2]. Taste disturbances start within 2 to 3 weeks after chemotherapy and can continue throughout treatment [1]. The literature on TA using subjective or objective analysis is limited, but it shows a high prevalence of TAs during chemotherapy ranges between 49.4% [3], and 76.1% [4]. Furthermore, TAs occur in at least one of the five basic tastes, with sweetness being the most affected [5], and patients exhibiting increased sensitivity to sweet taste, accompanied by a significant decrease in sweet thresholds [6]. However, some research found salty tastes to be more affected and difficult to taste than sweet tastes [2].

TA during chemotherapy have been identified as a serious problem [2], and more studies have investigated its effects patients' lifestyles and dietary habits. It has been shown that TA in cancer patients may influence their eating habits and appetite, leading to decreased body weight and possible deficiencies in essential nutrients [7]. In addition, TA may increase the risk of developing malnutrition in cancer patient. However, research on the precise correlation between TAs and malnutrition are limited. While certain studies have suggested that dietary habits may not have a direct relationship with taste alteration during chemotherapy [8], others have emphasized the significance of taste alteration as a side effect in cancer patients [9], which could potentially result in deficiencies of macro- and micronutrients [7].

There are numerous factors associated with TAs during chemotherapy. Smokers and older patients were less affected by chemotherapy-induced taste impairments due to their increased taste thresholds [2, 10]. On the other hand, women were found to be more susceptible to dysgeusia than men [11, 12]. Developing dysgeusia was significantly associated with the type of cancer and the chemotherapy regimen [13]. Lung and breast cancer patients were more likely to have TAs due to chemotherapy regimens employed [2]. Similarly, gynecological cancer patients also showed a greater incidence of TAs [4]. Furthermore, it was observed that the number of chemotherapy cycles were associated with TA [4, 11, 13, 14].

To cope with the TAs, patients applied several behavioral and self-management strategies. Examples included eating highly seasoned foods, experimenting with new recipes, catering to specific food cravings, cutting foods with lemon, eating sweets before meals, drinking sweetened beverages, drinking with a straw, and eating with plastic utensils; brushing teeth and tongue before eating; and using baking soda, salt, or antibacterial mouthwashes [15].

The aim of this study is to investigate the prevalence of taste alterations in cancer patients undergoing chemotherapy and their association with nutritional status, comorbid diseases, and malnutrition. Furthermore, the factors associated with TAs, copings strategies to manage TAs, and the prevalence of malnutrition is investigated.

## Methods

### Study design and population

The present study used a cross-sectional design. The sample size was estimated using the Coshrans' formula for cross-sectional studies to determine the prevalence of taste changes. The

prevalence of taste alteration was derived from a prior study conducted by Özkan et al. [16]. It was 63%, with an anticipated difference of 10%, an alpha of 0.05, and a power of 80%. The sample size was calculated to be 89 patients, but using the mean difference between two independent groups, an accepted margin of error of 5%, a confidence level of 95%, and a power of 80%. A total of 120 patients were required for the study.

Patients who were included in this study are cancer patients who are at least 18 years old, had chemotherapy at least once, capable of oral intake, and who can sign the consent form, while patients with chronic disease that may affect the taste (i.e., chronic kidney disease), has taste and smell altered prior to starting chemotherapy (i.e., COVID patients), and with cognitive impairment were excluded from the study.

## Data collection and research tools

Data was collected from December 1st, 2022, to March 31st, 2023. The study sample was recruited using convenience sampling technique from the oncology departments of three medical centers: An-Najah National University Hospital in Nablus, Al-Hussain Hospital in Beit Jala, and Palestine Medical Complex in Ramallah through face-to-face interviews.

A four-part structured interview was conducted during chemotherapy with subjects who met the inclusion criteria, agreed to participate in the study, and signed the informed consent. Each subject's name was recorded, and a code was assigned to him or her on the data sheet. The interview lasted between 25 and 35 minutes. The first section discusses sociodemographic and lifestyle. The second part focuses on cancer-related information, chemotherapy side effects, changes in taste perception, and the assessment of comorbid diseases. The third section includes nutritional status assessment, and the interview ended with questions about strategies patients might use to cope with TAs.

The study conducted reliability tests for the majority of instruments. Given that the tools were being employed for the first time in research conducted in Palestine, their content validity and dependability were evaluated using reliability testing. Significantly, every tool was either accessible in its Arabic version or had been translated into Arabic before being utilized. Furthermore, as each scale evaluates a distinct variable, reliability tests were performed separately for each scale.

**The antineoplastic side effects scale (ASES).**   The newly designed Arabic scale ASES was used to subjectively measure chemotherapy side effects. It has good validity and reliability with a Cronbach's alpha of 0.91. It evaluates 40 distinctive chemotherapy side effects and consists of three subscales: frequency, severity, and how the side effects affect patients' daily activities [17]. In this study, the reliability test for ASES resulted in Cronbach's alpha of 0.737 for the ASES frequency subscale, 0.876 for the severity subscale, and 0.896 for the daily life effect subscale, whereas the reliability of the total scale was 0.933.

**Malnutrition universal screening tool (MUST).**   Clinical assessment was performed using the Arabic version of the validated and reliable MUST (Cronbach's alpha = 0.79) [18], a validated tool for routine nutrition screening [19]. MUST is a five-step screening tool to identify adults who are malnourished or at risk of malnutrition [20].

**Chemotherapy-induced taste alteration scale (CiTAS).**   The subjective evaluation of taste changes was performed with the instrument CiTAS, which has excellent reliability (Cronbach alpha = 0.9) and good validity [21]. The original version of the CiTAS is an 18-item, self-administered questionnaire using a five-point Likert scale [22]. An increasing score indicates higher TA intensity [23]. The prevalence related to each subscale was determined by counting the number of patients with scores higher than 1 and calculating the sum as a percentage of the study population, and the overall prevalence of TAs was

calculated in the same way, as used by Larsen et al. [24]. In this study, CiTAS was used in the Arabic version after back-to-back translation, and the Cronbach's alpha was determined to be 0.883.

**Charlson comorbidities index (CCI).** The Charlson comorbidity index is a validated comorbidity assessment tool that can predict mortality in patients with various diseases. The original Charlson index used in this study included 17 comorbidities with dichotomous responses (yes and no). Among these, three comorbid conditions were mutually exclusive: diabetes with chronic complications and diabetes without chronic complications; mild liver disease and moderate or severe liver disease; and any malignancy and metastatic solid tumor [25]. Each comorbid disease is assigned for a weight of 1, 2, 3, or 6, and the CCI total score is determined by summing all weights [26].

**Nutritional status assessment.** Nutritional status was assessed using anthropometric measurements, biochemical data, and clinical assessment. Weight and height were obtained from patient records. The body mass index was calculated as body weight in kilograms divided by height squared in meters ($kg/m^2$). Biochemical data for albumin, hemoglobin, total protein, and C-reactive protein were obtained from patient recent records. These measurements were used to assess the patient's overall condition and nutritional status [27]. Clinical assessment was performed using the Arabic version of the mentioned earlier MUST.

**Coping-strategies evaluation.** A data sheet from a prior study was used to assess how patients tolerate and manage taste alterations throughout treatment [28]. It includes twenty tips for minimizing taste changes that can be adopted. The Cronbach's alpha for coping strategies in this study was 0.877.

## Statistical analysis

The data was analyzed using the Statistical Package for the Social Sciences (SPSS) version 21. The normality of the distribution of continuous variables was assessed graphically and using the Shapiro-Wilk Test. Continuous variables were analyzed using descriptive statistics such as means and standard deviation, while categorical variables were described using percentages and frequencies. To investigate the relationship between continuous and categorical variables, the one-way ANOVA or independent sample t-test was used, and the level of significance was set at $p < 0.05$.

## Ethics

Once a participant is identified to meet the inclusion criteria, they were handled the information sheet which included information about the study and a consent form. Participant were directed to contract the research term if they have any concern or questions before starting the interview. If they agree to participate, they were required to sign the consent form. All data were remained anonymous, and no name was associated with any data resulted from the study. All forms contained only an assigned number used in place of the subject's name in all field notes. All subject information were kept confidential and secure by locking field notes, data sheets, and consent forms in password protected files. The protocol for this study has approved by the institutional review board (IRB) ethical committee at An-Najah National University (Ref: Mas. Oct. 2022/40), while permissions and approval to conduct the study were obtained from the Palestinian Ministry of Health (Ref: 162/2454/2022) and the An-Najah National University Hospital administration.

# Results

## Patient characteristics

One hundred and twenty cancer patients took part in this study. Participants characteristics are shown in Table 1.

**Table 1. Patients' sociodemographic and lifestyle characteristics presented in n (%).**

| Characteristics (n = 120) | | n | % |
|---|---|---|---|
| Gender | Male | 29 | 24.2 |
| | Female | 91 | 75.8 |
| Age | 18–35 years | 24 | 20 |
| | 36–55 years | 54 | 45 |
| | 56–75 years | 40 | 33.3 |
| | Above 75 years | 2 | 1.7 |
| Marital status | Married | 99 | 82.5 |
| | Single | 18 | 15 |
| | Other | 3 | 2.5 |
| Educational level | No formal education | 6 | 5 |
| | Primary school | 46 | 38.3 |
| | Secondary school | 39 | 32.5 |
| | Diploma | 11 | 9.2 |
| | Postgraduate | 18 | 15 |
| Living area | City | 52 | 43.3 |
| | Village | 63 | 52.5 |
| | Camp | 5 | 4.2 |
| Living status | With spouse | 93 | 77.5 |
| | With family | 23 | 19.2 |
| | Alone | 2 | 1.7 |
| | Other | 2 | 1.7 |
| Working status | Full time | 20 | 16.7 |
| | Part time | 2 | 1.7 |
| | Unemployed | 95 | 79.2 |
| | Retired | 3 | 2.5 |
| Monthly income (NIS/month) | < 1500 | 18 | 15 |
| | 1500–3000 | 62 | 51.7 |
| | 3000–5000 | 31 | 25.8 |
| | > 5000 | 9 | 7.5 |
| Smoking status | Smoker | 13 | 10.8 |
| | Former smoker | 14 | 11.7 |
| | Nonsmoker | 93 | 77.5 |
| Sleep duration | < 6 hour/day | 21 | 17.5 |
| | 6–8 hour/day | 87 | 72.5 |
| | > 8 hour/day | 12 | 10 |
| Sleeping problem | Yes | 82 | 68.3 |
| | No | 38 | 31.7 |
| Physical activity compared to before CT | More active | 4 | 21.7 |
| | The same | 12 | 10 |
| | Less active | 104 | 86.7 |

NIS: New Israeli shekel.

## Medical history

Among participants, hypertension (27.5%) and diabetes (20%) were the most common comorbidities. Followed by asthma, mild liver disease, and rheumatic disease (Table 2). The mean score of the CCI was 4.15±2.31. The number of patients with CCI scores 1–2 (mildly ill), 3–4 (moderately ill), and ≥ 5 (severely ill) was 49 (40.8%), 25 (20.9%), and 50 (38.3%), respectively.

## Nutritional status assessment

**Anthropometric and clinical data.** According to BMI, 36.66% were obese, 35% were overweight, 27.5% were normal weight, and 0.83% were underweight. The mean weight before starting chemotherapy was 79.43±17.34 kg, and the mean current weight was 77.41±17.23 kg. The malnutrition universal screening tool revealed that 58 patients (48.3%) were at low risk of malnutrition, 24 (20%) were at medium risk, and 38 (31.7%) were at high risk of malnutrition (Fig 1).

**Biochemical data.** Biochemical tests revealed that most patients (60.83%) had low hemoglobin (11.46±1.68). Also, 60.83% of the patient's albumin levels were recorded, and most of them (89.04%) had normal levels (3.95±0.52) (Table 3).

## Cancer-related data

The most prevalent type of cancer was breast cancer (45.8%), then colon cancer (14.2%), lymphoma (14.2%), and lung cancer (8.3%). Among participants, 64.2% of the patients had no metastasis, and 16.7%, 12.5% were receiving paclitaxel, Adriamycin plus cyclophosphamide chemotherapy drugs, respectively. The mean number of finished chemotherapy sessions

**Table 2. Prevalence of comorbid conditions in Charlson comorbidity index presented in n (%).**

| Comorbid condition | Assigned weighing* | n (%) |
|---|---|---|
| Myocardial infraction | 1 | 1 (0.8) |
| Congestive heart failure | | 1 (0.8) |
| Peripheral vascular disease or bypass | | 0 |
| Cerebrovascular disease or transient ischemic disease | | 0 |
| Gastric peptic ulcer | | 0 |
| Pulmonary disease/ asthma | | 5 (4.2) |
| Diabetes | | 24 (20) |
| Dementia or Alzheimer's | | 0 |
| Rheumatic or connective tissue disease | | 3 (2.5) |
| Hypertension | | 33 (27.5) |
| Depression | | 0 |
| Warfarin | | 0 |
| Diabetes with end organ damage | 2 | 6 (5) |
| Cancer (lymphoma, leukemia, solid tumor) | | 77 (64.2) |
| Renal disease | | 1 (0.8) |
| Skin ulcer/ cellulitis | | 0 |
| Mild liver disease | | 3 (2.5) |
| Severe liver disease | 3 | 0 |
| Metastatic solid tumor | 6 | 43 (35.8) |
| HIV or AIDS | | 0 |

* Weighting of each variable as in the Charlson Comorbidity Index from 1–6, with a weight of six representing the most severe morbidity [29].

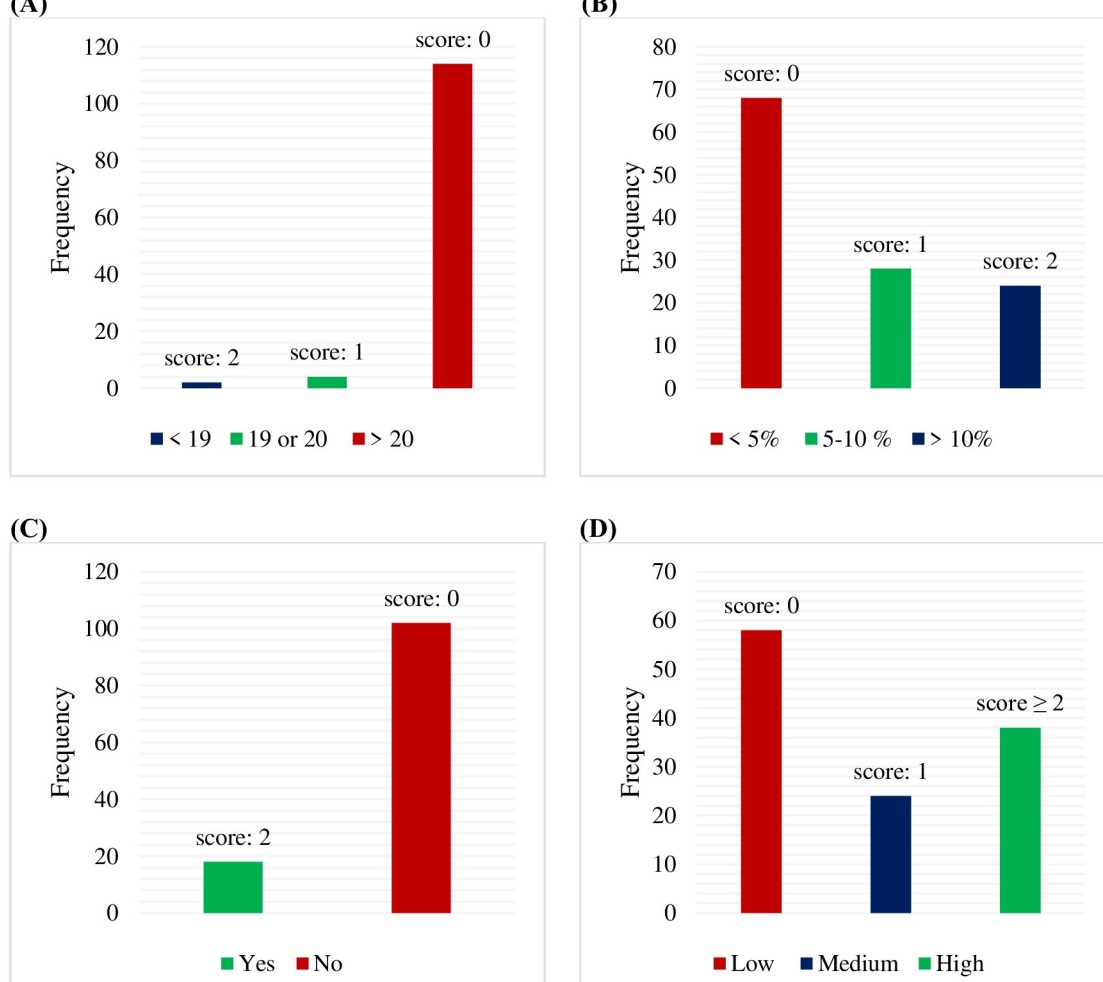

**Fig 1. MUST steps for malnutrition screening.** (A) BMI. (B) Weight loss percentage. (C) Acute disease effect. (D) Malnutrition risk.

Table 3. Patient's biochemical levels presented in n (%).

| Test (Normal value-lab report) | | n | % |
|---|---|---|---|
| Albumin (3.5–5.2 g/dl) | Low | 8 | 10.9 |
| n = 73 | Normal | 65 | 89.04 |
| | High | 0 | 0 |
| Hemoglobin (12–16 g/dl) | Low | 73 | 60.83 |
| n = 120 | Normal | 46 | 38.33 |
| | High | 1 | 0.8 |
| Total protein (6–8.3 g/dl) | Low | 6 | 14.28 |
| n = 42 | Normal | 34 | 80.95 |
| | High | 2 | 4.76 |
| C-reactive protein (0.8–1 mg/dl and lower) | Normal | 4 | 10.25 |
| n = 39 | High | 35 | 89.74 |

received by the patients was 5.83±7.9, and 20%, 18.3% of the patients received one chemotherapy cycle and three chemotherapy cycles, respectively (Table 4).

Table 5 shows the complete list of the 40 side effects listed in the ASES. Increased or poor appetite (91.6%), lack of energy (89.2%), generalized pain (79.2%), dry mouth (76.6%), hair loss (72.5%), nausea (70.8%), and changes in how things smell, or taste (70.8%) were the most common side effects.

**Table 4. Cancer-related characteristics presented in n (%).**

| Characteristics (n = 120) | | n | % |
|---|---|---|---|
| Cancer diagnosis | Breast | 55 | 45.8 |
| | Colon | 17 | 14.2 |
| | Lymphoma | 17 | 14.2 |
| | Lung | 10 | 8.3 |
| | Other | 21 | 17.4 |
| Metastasis | Yes | 43 | 35.8 |
| | No | 77 | 64.2 |
| Type of therapy | Chemotherapy | 120 | 100 |
| | Radiation | 11 | 9.2 |
| | Biological/hormonal | 36 | 30 |
| | Surgical | 32 | 26.7 |
| Chemotherapy protocol/ drug | Paclitaxel | 20 | 16.7 |
| | AC | 15 | 12.5 |
| | ABVD | 8 | 6.7 |
| | Gemcitabine | 8 | 6.7 |
| | Xelox | 8 | 6.7 |
| | Docetaxel | 6 | 5 |
| | PC | 6 | 5 |
| | Gemcitabine/Cisplatin | 4 | 3.3 |
| | XELIRI | 4 | 3.3 |
| | FOLFOX | 3 | 2.5 |
| | Pemetrexed/Carboplatin | 3 | 2.5 |
| | Arsenic Trioxide/Tretinoin | 3 | 2.5 |
| | Docetaxel/Carboplatin | 3 | 2.5 |
| | Gemcitabine/Carboplatin | 2 | 1.7 |
| | BEACOPP | 2 | 1.7 |
| | EC | 2 | 1.7 |
| | Ifosfamide/Gemcitabine/Vinorelbine | 2 | 1.7 |
| | Capecitabine | 2 | 1.7 |
| | Other | 19 | 15.8 |
| Number of finished chemotherapy cycles | 1 | 24 | 20 |
| | 2 | 16 | 13.3 |
| | 3 | 22 | 18.3 |
| | 4 | 12 | 10 |
| | 5 and over | 46 | 38.3 |

AC: Adriamycin (doxorubicin hydrochloride), Cyclophosphamide. ABVD: Adriamycin, Bleomycin, Vinblastine, Dacarbazine. BEACOPP: Bleomycin sulfate, Etoposide phosphate, Doxorubicin hydrochloride (Adriamycin), Cyclophosphamide, Vincristine sulfate (Oncovin), Procarbazine hydrochloride, and Prednisone. EC: Epirubicin, Cyclophosphamide. PC: Paclitaxel, Carboplatin. FOLFOX: Oxaliplatin, 5-Fluorouracil, Leucovorin. Xelox: Oxaliplatin, Capecitabine. XELIRI: Irinotecan, Capecitabine.

**Table 5. Chemotherapy side effect frequency, severity, and daily activity effect presented in n (%) and mean (SD).**

| Side effect | Frequency | Severity | Effect on daily activity (Mean ± SD) |
|---|---|---|---|
| | n (%) | (Mean ± SD) | |
| Increased or poor appetite | 110 (91.6) | 6.85 ± 2.43 | 2.54 ± 1.22 |
| Lack of energy | 107 (89.2) | 6.74 ± 2.64 | 2.98 ± 1.41 |
| Generalized pain | 95 (79.2) | 5.88 ± 3.30 | 2.61 ± 1.66 |
| Dry mouth | 92 (76.6) | 5.78 ± 3.53 | 2.19 ± 1.50 |
| Hair loss | 87 (72.5) | 6.08 ± 4.15 | 2.18 ± 1.87 |
| Changes in how things smell or taste | 85 (70.8) | 5.40 ± 3.86 | 2.38 ± 1.84 |
| Nausea | 85 (70.8) | 5.01 ± 3.60 | 1.85 ± 1.51 |
| Painful/ increased urination | 83 (69.2) | 4.72 ± 3.54 | 1.82 ± 1.50 |
| Anxiety | 82 (68.3) | 4.71 ± 3.48 | 1.85 ± 1.52 |
| Difficulty remembering things | 77 (64.2) | 4.42 ± 3.70 | 1.92 ± 1.72 |
| Disturbed sleep | 77 (64.2) | 4.92 ± 3.93 | 2.07 ± 1.81 |
| Dizziness | 76 (63.3) | 4.36 ± 3.43 | 1.81 ± 1.54 |
| Feeling angry | 74 (61.6) | 3.93 ± 3.59 | 1.70 ± 1.60 |
| Feeling nervous | 73 (60.8) | 3.86 ± 3.43 | 1.58 ± 1.53 |
| Feeling bloated | 73 (60.8) | 4.04 ± 3.52 | 1.46 ± 1.42 |
| Numbness and tingling sensation in feet or hand | 71 (59.2) | 3.98 ± 3.55 | 1.44 ± 1.43 |
| Abdominal pain | 70 (58.3) | 3.69 ± 3.43 | 1.37 ± 1.50 |
| Difficulty concentrating | 68 (56.6) | 3.96 ± 3.68 | 1.66 ± 1.64 |
| Crying more often | 67 (55.8) | 3.67 ± 3.65 | 1.32 ± 1.42 |
| Excessive thirst | 66 (55) | 4.11 ± 3.94 | 1.38 ± 1.59 |
| Dry skin | 65 (54.2) | 3.69 ± 3.57 | 1.01 ± 1.13 |
| Weight loss | 61 (50.8) | 3.10 ± 3.30 | 0.88 ± 1.05 |
| Constipation | 60 (50) | 3.28 ± 3.52 | 1.27 ± 1.45 |
| Feeling sad or depressed | 58 (48.3) | 2.87 ± 3.30 | 1.23 ± 1.47 |
| Palpitation | 58 (48.3) | 3.06 ± 3.38 | 1.21 ± 1.41 |
| Sour mouth or throat | 58 (48.3) | 3.3 ± 3.62 | 1.28 ± 1.52 |
| Changes in skin color | 58 (48.3) | 3.26 ± 3.61 | 0.97 ± 1.26 |
| Diarrhea | 53 (44.2) | 2.69 ± 3.30 | 1.07 ± 1.39 |
| Fear | 51 (42.5) | 2.58 ± 3.37 | 0.98 ± 1.33 |
| Weight gain | 50 (41.6) | 2.29 ± 2.92 | 0.73 ± 1.07 |
| Easily bruising | 47 (39.2) | 2.31 ± 3.15 | 0.62 ± 0.88 |
| Shortness of breath | 46 (38.3) | 2.45 ± 3.38 | 0.92 ± 1.33 |
| Itching | 46 (38.3) | 2.43 ± 3.26 | 0.79 ± 1.15 |
| Vomiting | 43 (35.8) | 2.38 ± 3.43 | 0.88 ±1.38 |
| Confusion | 38 (31.6) | 2.00 ± 3.12 | 0.76 ± 1.18 |
| Difficulty swallowing | 31 (25.8) | 1.69 ± 2.97 | 0.68 ± 1.27 |
| Problem with sexual interest or activity | 22 (18.3) | 1.32 ± 2.93 | 0.32 ± 0.77 |
| Skin rash | 19 (15.8) | 0.98 ± 2.39 | 0.33 ± 0.82 |
| Acne | 15 (12.5) | 0.65 ± 1.89 | 0.21 ± 0.69 |
| Excessive hair growth | 5 (4.2) | 0.18 ± 0.96 | 0.04 ± 0.20 |

When the severity subscale of the ASES was considered, increased or poor appetite appeared to be the most acutely perceived side effect with a score of 6.85±2.43 on a scale ranging from 1 to 10, followed by lack of energy, hair loss, generalized pain, dry mouth, changes in how things smell or taste, and nausea with scores of 6.74±2.64, 6.08±4.15, 5.88±3.30, 5.78

±3.53, 5.40±3.86, and 5.01±3.60, respectively. Acne (0.65±1.89) and excessive hair growth (0.18±0.96) were the least annoying side effects.

On the ASES subscale describing the impact of side effects on activities of daily living, the highest scores on a scale ranging from 1 to 5 were lack of energy 2.98±1.41, generalized pain 2.61±1.66, increased or poor appetite 2.54±1.22, changes in how things smell or taste 2.38 ±1.84.

## Taste alteration-related data

The mean scores of patients from the subscales of the chemotherapy-induced taste alterations scale were as follows: decrease in basic taste 1.81±1.37, discomfort 2.72±0.97, phantogeusia and parageusia 2.4±1.32, and general taste changes 2.66±1.42 (Table 6). With a CiTAS score ranging from 1 to 5, chemotherapy-induced taste changes can be classified as moderate.

Most patients did not try any of the proposed self-management strategies to cope with taste alteration (Table 7). Eating more flavored protein food was the most useful method for 14.2% of the patients to cope with taste alterations, followed by 10.8% for eating more bland food, boiling food to make it blander, and eating cold food. Brushing one's teeth before eating was the least useful advice attempted, while using plastic silverware was the least tried.

## Prevalence of taste alteration

The incidence of taste alteration acquired from self-reported taste and smell changes on the antineoplastic side effect scale was 70.8%. The prevalence of overall taste alterations was 98.3%, according to CiTAS. Regarding the CiTAS subscales, 32% of participants reported a reduction

**Table 6. Cancer-induced taste alteration scale score presented in n (%) and mean (SD).**

|  |  | n (%) | Mean ± SD |
|---|---|---|---|
| Taste changes subscales scores | Decline in basic taste | 39 (32.5) | 1.81±1.37 |
|  | Discomfort | 115 (95.8) | 2.72±0.97 |
|  | Parageusia and Phantogeusia | 81 (67.5) | 2.40±1.32 |
|  | General taste alterations | 88 (73.3) | 2.66±1.42 |
| Characteristic | Have difficulty tasting food | 78 (65) | 3.07±1.64 |
|  | Have difficulty tasting sweetness | 33 (27.5) | 1.85±1.47 |
|  | Have difficulty tasting saltiness | 34 (28.3) | 1.87±1.46 |
|  | Have difficulty tasting sourness | 34 (28.3) | 1.85±1.44 |
|  | Have difficulty tasting bitterness | 30 (25) | 1.77±1.40 |
|  | Have difficulty tasting umami | 29 (24.2) | 1.74±1.39 |
|  | Unable to perceive the smell or flavor of food | 52 (43.3) | 2.27±1.57 |
|  | Everything tastes bad | 65 (54.2) | 2.61±1.64 |
|  | Food doesn't taste as it should | 65 (54.2) | 2.73±1.72 |
|  | Have a bitter taste in the mouth | 71 (59.2) | 2.75±1.61 |
|  | Have a bad taste in the mouth | 56 (46.7) | 2.47±1.67 |
|  | Everything tastes bitter | 43 (35.8) | 1.98±1.43 |
|  | Feel nauseated and queasy | 82 (68.3) | 3.04±1.55 |
|  | Bothered by the smell of food | 65 (54.2) | 2.59±1.59 |
|  | Have difficulty eating hot food | 44 (36.7) | 2.01±1.45 |
|  | Have difficulty eating oily food | 60 (50) | 2.52±1.69 |
|  | Have difficulty eating meat | 61 (50.8) | 2.56±1.67 |
|  | Have a reduced appetite | 95 (79.2) | 3.64±1.55 |

**Table 7. Coping and self-management strategies to deal with taste alterations presented in n (%).**

| Suggestion | n = 120 | | | |
|---|---|---|---|---|
| | Did not try | Tried but did not help | Helped a little | Helped a lot |
| Increase seasonings or spices (oregano, basil, cinnamon, ginger) | 101 (84.2) | 7 (5.8) | 7 (5.8) | 5 (4.2) |
| Decrease seasoning or spices | 102 (85) | 9 (7.5) | 2 (1.7) | 7 (5.8) |
| Eat more bland foods | 92 (76.7) | 8 (6.7) | 7 (5.8) | 13 (10.8) |
| Boil food to make them blander | 95 (79.2) | 4 (3.3) | 8 (6.7) | 13 (10.8) |
| Use more salt | 102 (85) | 7 (5.8) | 5 (4.2) | 6 (5) |
| Use less salt | 108 (90) | 5 (4.2) | 4 (3.3) | 3 (2.5) |
| Use more condiments (mustard, ketchup, pickle, relish, hot peppers) | 89 (74.2) | 5 (4.2) | 15 (12.5) | 11 (9.2) |
| Add fats or sauces to food (gravy, butter, sour cream) | 113 (94.2) | 2 (1.7) | 4 (3.3) | 1 (0.8) |
| Eat foods at room temperature | 100 (83.3) | 5 (4.2) | 11 (9.2) | 4 (3.3) |
| Eat cold foods | 95 (79.2) | 2 (1.7) | 10 (8.3) | 13 (10.8) |
| Add something sweet with meats (cranberry, sauce, applesauce) | 115 (95.8) | 3 (2.5) | 2 (1.7) | 0 |
| Avoid beef | 107 (89.2) | 4 (3.3) | 5 (4.2) | 4 (3.3) |
| Avoid food with strong smells (fish) | 94 (78.3) | 7 (5.8) | 9 (7.5) | 10 (8.3) |
| Eat more protein food that have been flavored (eggs, beans, chicken) | 81 (67.5) | 5 (4.2) | 17 (14.2) | 17 (14.2) |
| Drink more water with food to help with eating or rinse away bad taste | 98 (81.7) | 1 (0.8) | 11 (9.2) | 10 (8.3) |
| Eat smaller, more frequent meals | 99 (82.5) | 2 (1.7) | 8 (6.7) | 11 (9.2) |
| Brush your teeth before eating | 98 (81.7) | 10 (8.3) | 6 (5) | 6 (5) |
| Suck on hard candy | 99 (82.5) | 3 (2.5) | 7 (5.8) | 11 (9.2) |
| Use plastic silverware | 117 (97.5) | 1 (0.8) | 0 | 2 (1.7) |
| Marinate meats to change taste | 105 (87.5) | 5 (4.2) | 7 (5.8) | 3 (2.5) |

in basic tastes, 95.8% reported oral discomfort, 67.5% reported phantogeusia and parageusia, and 73.4% reported general taste alterations (Fig 2).

## Taste alteration and nutritional status

There was no significant association between the risk of malnutrition and the subscales of basic taste reduction, phantogeusia and parageusia, as well as general taste alterations (p>0.05), but the second subscale (discomfort) was significantly associated with the risk of malnutrition (p<0.05).

The subscales of MUST: recent weight loss and acute disease effect were significantly associated with the chemotherapy-induced taste changes subscale (discomfort) (p<0.05) (Table 8). In addition, there was no statistical significance between the CiTAS subscales and BMI or biochemical data (p>0.05).

## Taste alteration and sociodemographic characteristic and lifestyle

Age was significantly associated with taste alteration-induced discomfort, phantogeusia and parageusia, as well as general taste alteration subscales (p<0.05). According to gender, the phantogeusia and parageusia subscale was significantly associated with gender (p<0.05). Educational level was found to be associated with the discomfort subscale (p<0.05) (Table 9).

CiTAS scores did not vary according to monthly income and smoking status (p>0.05). But sleeping problems were found to be associated with discomfort, phantogeusia and parageusia, as well as the general taste alterations subscale (p<0.05). However, patients' physical activity was significantly associated with phantogeusia and parageusia, as well as general taste alterations (p<0.05) (Table 9).

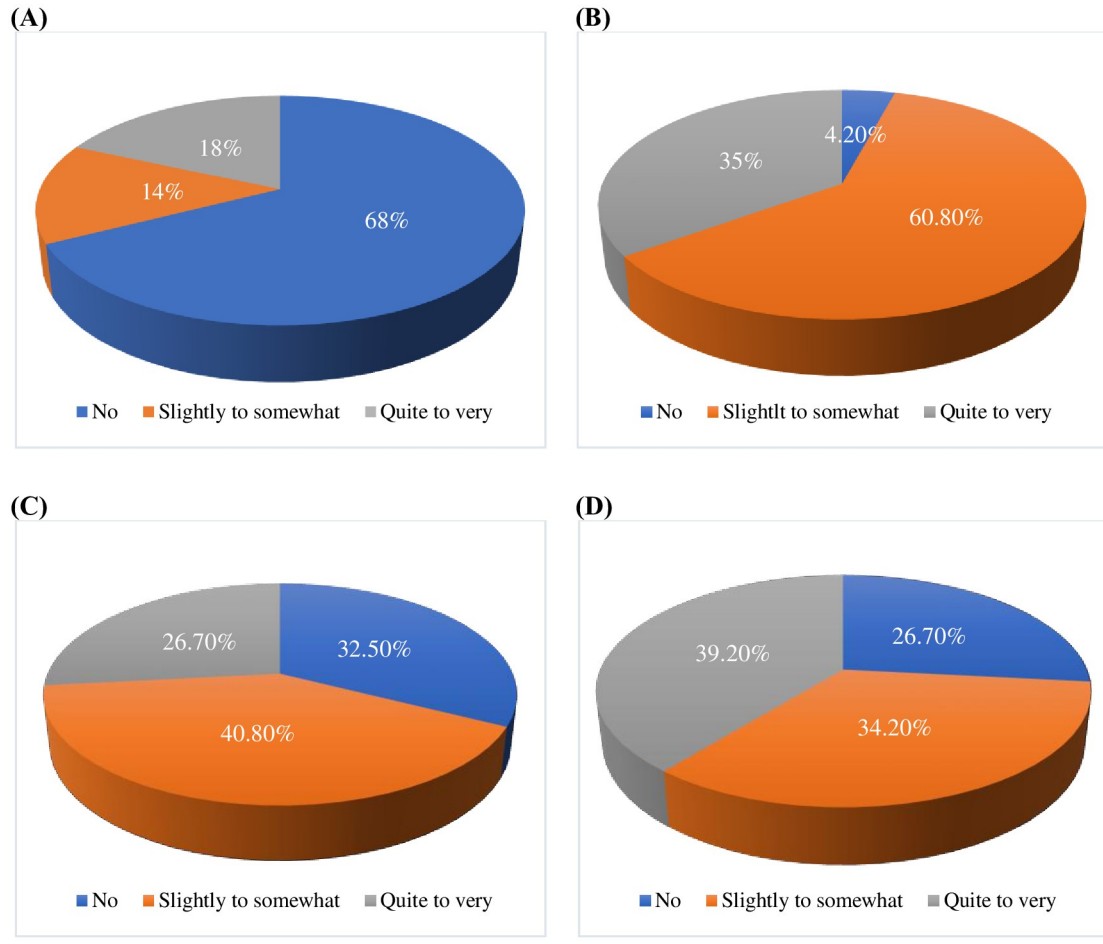

**Fig 2. Prevalence of taste alteration according to CiTAS subscale.** (A) Basic taste reduction subscale. (B) Taste disorder (discomfort) subscale. (C) Phantogeusia and parageusia subscale. (D) General taste alterations subscale.

**Table 8. Changes in CiTAS according to nutritional status factors measured by malnutrition universal screening tool MUST.**

| Variables n = 120 | | Decline in basic taste | Discomfort (Taste disorder) | Phantogeusia and parageusia | General TAs |
|---|---|---|---|---|---|
| | | Mean ± SD | Mean ± SD | Mean ± SD | Mean ± SD |
| Malnutrition risk | Low | 1.88±1.42 | 2.42±0.78 | 2.33±1.28 | 2.58±1.41 |
| | Medium | 1.52±1.17 | 2.82±1.14 | 2.09±1.19 | 2.35±1.40 |
| | High | 1.88±1.41 | 3.12±1.00 | 2.68±1.43 | 2.99±1.41 |
| | P-value | 0.517 | 0.002* | 0.211 | 0.187 |
| Recent weight loss % | < 5% | 1.87±1.41 | 2.48±0.82 | 2.36±1.26 | 2.61±1.40 |
| | 5–10% | 1.78±1.42 | 2.94±1.08 | 2.27±1.43 | 2.43±1.51 |
| | > 10% | 1.67±1.23 | 3.15±1.08 | 2.63±1.38 | 3.08±1.33 |
| | P-value | 0.822 | 0.005* | 0.588 | 0.239 |
| Acute disease effect | Yes | 2.00±1.54 | 3.33±0.79 | 2.87±1.41 | 3.15±1.40 |
| | No | 1.78±1.34 | 2.61±0.97 | 2.31±1.29 | 2.58±1.41 |
| | P-value | 0.538 | 0.004* | 0.103 | 0.116 |

Significant at *: p<0.05 according to one-way ANOVA/independent t-test.

**Table 9. Effect of sociodemographic factors and lifestyle on changes in CiTAS.**

| Variables n = 120 | | Decline in basic taste | Discomfort (Taste disorder) | Phantogeusia and parageusia | General TAs |
|---|---|---|---|---|---|
| | | Mean ± SD | Mean ±SD | Mean ± SD | Mean ± SD |
| Age | 18–35 years | 1.02±0.12 | 2.34±0.68 | 2.01±1.27 | 1.83±1.18 |
| | 36–55 years | 2.31±1.59 | 2.60±1.03 | 2.64±1.33 | 3.08±1.42 |
| | 56–75 years | 1.65±1.23 | 3.05±0.93 | 2.35±1.30 | 2.61±1.37 |
| | > 75 years | 1.00±0.00 | 4.08±0.11 | 1.33±0.47 | 2.37±0.17 |
| | P-value | 0.001* | 0.004* | 0.152 | 0.004* |
| Gender | Male | 1.68±1.28 | 2.48±0.80 | 1.94±1.18 | 2.30±1.47 |
| | Female | 1.85±1.40 | 2.80±1.01 | 2.54±1.34 | 2.78±1.39 |
| | P-value | 0.575 | 0.124 | 0.032* | 0.113 |
| Educational level | No school | 2.13±1.47 | 4.19±0.74 | 3.66±1.05 | 3.41±0.64 |
| | Primary School | 2.02±1.46 | 2.73±0.90 | 2.50±1.30 | 2.84±1.40 |
| | Secondary school | 1.87±1.48 | 2.51±0.99 | 2.08±1.23 | 2.51±1.47 |
| | Diploma | 1.45±1.19 | 2.68±0.70 | 2.60±1.25 | 2.22±1.33 |
| | Postgrad. | 1.27±0.76 | 2.68±1.02 | 2.27±1.49 | 2.54±1.56 |
| | P-value | 0.297 | 0.003* | 0.077 | 0.409 |
| Monthly income | < 1500 | 1.85±1.34 | 2.72±1.04 | 2.35±1.27 | 2.73±1.62 |
| | 1500–5000 | 1.81±1.37 | 2.72±0.97 | 2.40±1.34 | 2.63±1.37 |
| | > 5000 | 1.77±1.56 | 2.77±0.93 | 2.40±1.39 | 2.86±1.64 |
| | P-value | 0.989 | 0.987 | 0.986 | 0.881 |
| Smoking status | Smoker | 2.00±1.63 | 2.25±0.90 | 2.25±1.26 | 2.17±1.40 |
| | Former smoker | 1.71±1.13 | 2.55±0.81 | 2.14±1.24 | 2.60±1.38 |
| | Nonsmoker | 1.80±1.37 | 2.81±0.99 | 2.45±1.35 | 2.74±1.43 |
| | P-value | 0.856 | 0.122 | 0.653 | 0.397 |
| Sleeping issue | Yes | 1.94±1.48 | 2.86±1.02 | 2.58±1.35 | 2.95±1.42 |
| | No | 1.52±1.07 | 2.41±0.79 | 2.00±1.18 | 2.03±1.22 |
| | P-value | 0.118 | 0.018* | 0.024* | 0.001* |
| Physical activity | More active | 1.30±0.60 | 2.33±1.35 | 2.25±1.89 | 2.06±1.66 |
| | The same | 1.51±1.20 | 2.36±0.81 | 1.36±0.65 | 1.47±0.90 |
| | Less active | 1.86±1.41 | 2.78±0.97 | 2.52±1.31 | 2.82±1.40 |
| | P-value | 0.529 | 0.265 | 0.014* | 0.005* |

Significant at *: p<0.05, **: p<0.001 according to one-way ANOVA/independent t-test.

## Taste alteration and cancer-related data, antineoplastic side effects, and comorbidities

The phantogeusia and parageusia CiTAS subscale score was varied based on the type of cancer (p<0.05). The CiTAS subscales scores did not vary between patients with and without metastasis (p>0.05). According to type of chemotherapy, taxane-based chemotherapy was significantly associated with phantogeusia and parageusia, as well as general taste alteration subscales (p<0.05). Adriamycin and cyclophosphamide chemotherapy were also associated with the phantogeusia and parageusia subscale (p<0.05). However, CiTAS scores did not vary with platinum-based chemotherapy, gemcitabine, bleomycin, etoposide, or capecitabine chemotherapy regimens (p>0.05). Otherwise, basic taste reduction, phantogeusia and parageusia CiTAS subscales were significantly associated with the number of finished chemotherapy cycles (P<0.05) (Table 10).

**Table 10. Effect of cancer-related data on taste alterations.**

| Variables n = 120 | | Decline in basic taste | Discomfort (Taste disorder) | Phantogeusia and parageusia | General TAs |
|---|---|---|---|---|---|
| | | Mean ± SD | Mean ±SD | Mean ± SD | Mean ± SD |
| Type of cancer | Breast | 2.02±1.52 | 2.60±1.00 | 2.81±1.37 | 2.97±1.41 |
| | Colon | 1.82±1.42 | 2.76±0.93 | 1.92±1.11 | 2.58±1.34 |
| | Lymphoma | 1.47±1.17 | 2.39±0.92 | 2.15±1.31 | 2.04±1.31 |
| | Lung | 1.40±0.84 | 2.83±0.79 | 1.83±1.12 | 2.35±1.07 |
| | Other | 1.74±1.25 | 3.23±0.95 | 2.15±1.21 | 2.58±1.62 |
| | P-value | 0.518 | 0.071 | 0.026* | 0.170 |
| Metastasis | Yes | 1.83±1.32 | 2.84±0.88 | 2.27±1.20 | 2.69±1.24 |
| | No | 1.80±1.41 | 2.65±1.02 | 2.46±1.39 | 2.64±1.52 |
| | P-value | 0.895 | 0.308 | 0.457 | 0.859 |
| Taxane CT | Yes | 2.07±1.55 | 2.78±1.07 | 2.80±1.38 | 3.16±1.53 |
| | No | 1.79±1.27 | 2.69±0.93 | 2.21±1.26 | 2.43±1.31 |
| | P-value | 0.162 | 0.657 | 0.021* | 0.009* |
| Platinum CT | Yes | 2.16±1.44 | 2.91±0.98 | 2.41±1.17 | 2.72±1.30 |
| | No | 1.70±1.34 | 2.66±0.97 | 2.39±1.37 | 2.64±1.46 |
| | P-value | 0.113 | 0.220 | 0.958 | 0.797 |
| Adriamycin CT | Yes | 1.82±1.46 | 2.58±0.97 | 2.90±1.37 | 2.75±1.34 |
| | No | 1.81±1.35 | 2.76±0.97 | 2.25±1.28 | 2.63±1.45 |
| | P-value | 0.950 | 0.401 | 0.025* | 0.703 |
| Cyclophosph-amide CT | Yes | 2.18±1.61 | 2.39±1.01 | 3.15±1.30 | 3.01±1.39 |
| | No | 1.72±1.29 | 2.80±0.95 | 2.21±1.26 | 2.58±1.42 |
| | P-value | 0.143 | 0.0.64 | 0.002* | 0.187 |
| Gemcitabine CT | Yes | 1.57±1.22 | 2.74±0.97 | 1.88±1.12 | 2.20±1.29 |
| | No | 1.85±1.39 | 2.72±0.98 | 2.48±1.34 | 2.74±1.43 |
| | P-value | 0.442 | 0.933 | 0.082 | 0.151 |
| Bleomycin CT | Yes | 1.09±0.30 | 2.51±1.02 | 2.18±1.09 | 1.90±1.06 |
| | No | 1.88±1.41 | 2.74±0.97 | 2.42±1.34 | 2.74±1.43 |
| | P-value | 0.067 | 0.455 | 0.569 | 0.64 |
| Etoposide CT | Yes | 2.00±1.26 | 2.16±0.66 | 2.22±1.37 | 2.12±0.84 |
| | No | 1.80±1.38 | 2.75±0.98 | 2.40±1.32 | 2.69±1.44 |
| | P-value | 0.737 | 0.151 | 0.738 | 0.341 |
| Capecitabine CT | Yes | 1.71±1.43 | 2.71±0.94 | 1.78±0.99 | 2.41±1.36 |
| | No | 1.83±1.37 | 2.73±0.98 | 2.48±1.35 | 2.71±1.43 |
| | P-value | 0.757 | 0.941 | 0.66 | 0.452 |
| Number of finished CT cycle | 1 cycle | 1.55±1.09 | 2.66±0.95 | 2.30±1.17 | 2.44±1.44 |
| | 2 cycles | 1.88±1.34 | 2.96±0.99 | 2.02±1.15 | 2.75±1.37 |
| | 3 cycles | 2.65±1.62 | 2.83±1.05 | 2.95±1.29 | 2.88±1.24 |
| | 4 cycles | 1.33±1.15 | 2.84±1.03 | 1.80±1.25 | 2.00±1.28 |
| | 5 cycles | 1.22±0.52 | 2.75±1.25 | 3.50±1.44 | 3.25±1.44 |
| | ≥ 6 cycles | 1.76±1.45 | 2.54±0.87 | 2.18±1.29 | 2.70±1.54 |
| | P-value | 0.025* | 0.747 | 0.006* | 0.364 |

Significant at *: p<0.05 according to one-way ANOVA test/independent t-test.

Taste alteration was associated with some selected side effects and varied significantly concerning the severity of the side effects among the four CiTAS subscales. The decline in basic tastes CiTAS subscale was significantly associated with changes in how things smell or taste (p<0.05). The second subscale taste disorder (discomfort) varied significantly with discomfort,

disturbed sleep, dry mouth, sore mouth and throat, nausea, vomiting, appetite changes, difficulty swallowing, anxiety, weight loss, excessive thirst, and lack of energy ($p < 0.05$). Also, changes in how things smell or taste, dizziness, disturbed sleep, dry mouth, sore mouth and throat, nausea, weight loss, anxiety, excessive thirst, and difficulty concentrating were significantly associated with phantogeusia and parageusia ($p < 0.05$). The general taste alteration subscale was found to be significantly associated with changes in how things smell or taste, dizziness, disturbed sleep, dry mouth, sore mouth or throat, nausea, weight loss, excessive thirst, difficulty concentrating, and lack of energy ($p < 0.05$) (Table 11).

The comorbidities score was found to be associated with discomfort on the CiTAS subscale ($p < 0.05$). Patients with pulmonary disease were more associated with basic taste reduction ($p < 0.05$), as were patients with diabetes mellitus, which was found to be significantly associated with the taste alteration discomfort subscale ($p < 0.05$). Unlike diabetes mellitus (DM) with end organs, which has no statistical significance with CiTAS subscales ($p > 0.05$). Hypertension, however, was found to be significantly associated with the discomfort subscale ($p < 0.05$) (Table 12).

## Discussion

### Prevalence of taste alterations

Our study showed a high prevalence of self-reported TAs, up to 98.3% which is consistent with previous studies that indicated TAs can occur in up to 49.4% [3], 63.1% [16], 64% [13], 69.9% [10], 76.1% [4], and 93% of cancer patients taking chemotherapy using subjective assessment [24]. According to these findings, the difference in TAs prevalence, including our result, was thought to be due to sample size variation, study methodology, and the type of cancer analyzed in each study. However, several mechanisms have been proposed to clarify the phenomenon of TA regarding chemotherapy. It was hypothesized that TAs induced by chemotherapy were a result of a decrease in the count of receptor cells or impairments in neurotransmission. Taste receptor cells exhibit a brief lifespan and a rapid turnover rate of approximately 10 days, making them susceptible to chemotherapeutic agents that disrupt the metabolic processes of both healthy and malignant cells. Chemotherapy administration leads to the destruction of taste receptor cells, causing TAs that manifest shortly thereafter. However, upon discontinuation of medication, the taste alterations recover. The process of neurotransmission can be impacted in an indirect manner where cranial nerves are destroyed, afferent pathways are modified due to the passage of cytotoxic drugs through the blood-brain barrier, or as a consequence of neuropathy induced by chemotherapy, according to previous research [30]. In an alternative scenario, taste perception may be modified in an indirect manner as a result of chemotherapy-induced impairment of brain regions that regulate taste perception [11].

### Taste alterations and nutritional status

In this study, a statistically significant link was found between taste disorders and the risk of malnutrition. According to our results, a higher mean score on the taste disorder subscale suggested a higher risk of malnutrition. The taste disorder or discomfort subscale assesses the link between changes in taste sensation and nausea or vomiting, changes in the sense of smell, difficulty eating hot food, fatty food, meat, and appetite loss [12]. As a result, taste problems were assumed to have a serious effect on eating behavior and food selection, raising the risk of malnutrition, and making taste disorder an important determinant of malnutrition. Our study found no association between malnutrition risk and basic taste reduction, phantogeusia, parageusia, and general taste alteration, which may be attributable to sample size or malnutrition

**Table 11. Changes in Ci-TAS according to chemotherapeutic side effects.**

| Variables n = 120 | | Decline in basic taste | Discomfort (Taste disorder) | Phantogeusia and parageusia | General TAs |
|---|---|---|---|---|---|
| | | Mean ± SD | Mean ± SD | Mean ± SD | Mean ± SD |
| Changes in how things smell or taste | Not exist | 1.00±0.00 | 2.53±1.00 | 1.54±0.81 | 1.32±0.67 |
| | Moderate | 1.40±0.76 | 3.09±0.91 | 2.14±1.28 | 1.92±1.16 |
| | Severe | 2.23±1.55 | 2.78±0.96 | 2.82±1.33 | 3.36±1.21 |
| | P-value | 0.000** | 0.265 | 0.000** | 0.000** |
| Dizziness | Not exist | 1.78±1.44 | 2.44±0.95 | 1.81±1.20 | 2.17±1.44 |
| | Moderate | 1.47±0.88 | 2.72±0.90 | 2.56±1.03 | 2.46±1.32 |
| | Severe | 1.92±1.42 | 2.91±0.98 | 2.75±1.34 | 3.05±1.33 |
| | P-value | 0.506 | 0.049* | 0.001* | 0.006* |
| Disturbed sleep | Not exist | 1.63±1.26 | 2.38±0.77 | 1.89±1.20 | 1.97±1.28 |
| | Moderate | 1.83±1.17 | 2.88±0.47 | 2.38±1.49 | 2.66±1.20 |
| | Severe | 1.92±1.45 | 2.92±1.06 | 2.70±1.30 | 3.08±1.37 |
| | P-value | 0.552 | 0.015* | 0.005* | 0.000** |
| Dry mouth | Not exist | 1.72±1.36 | 2.31±0.87 | 1.69±0.92 | 2.13±1.33 |
| | Moderate | 2.27±1.50 | 2.58±1.00 | 2.12±1.35 | 2.46±1.52 |
| | Severe | 1.77±1.35 | 2.89±0.97 | 2.69±1.35 | 2.88±1.39 |
| | P-value | 0.437 | 0.022* | 0.002* | 0.046* |
| Sore mouth or throat | Not exist | 1.62±1.21 | 2.42±0.82 | 1.97±1.01 | 2.34±1.38 |
| | Moderate | 1.74±1.47 | 2.45±1.04 | 2.23±1.69 | 2.25±1.51 |
| | Severe | 2.08±1.52 | 3.18±0.99 | 3.00±1.40 | 3.18±1.32 |
| | P-value | 0.209 | 0.000** | 0.000** | 0.005* |
| Nausea | Not exist | 1.77±1.47 | 2.31±0.87 | 1.77±1.09 | 2.17±1.47 |
| | Moderate | 1.71±1.30 | 2.55±0.87 | 2.52±1.53 | 2.36±1.57 |
| | Severe | 1.86±1.35 | 2.97±0.98 | 2.69±1.28 | 2.99±1.27 |
| | P-value | 0.907 | 0.003* | 0.003* | 0.012* |
| Vomiting | Not exist | 1.89±1.46 | 2.56±0.93 | 2.34±1.30 | 2.58±1.45 |
| | Moderate | 1.21±0.60 | 2.46±0.72 | 2.60±1.43 | 2.34±1.49 |
| | Severe | 1.83±1.32 | 3.19±1.02 | 2.46±1.37 | 2.98±1.30 |
| | P-value | 0.317 | 0.006* | 0.782 | 0.296 |
| Increased or poor appetite | Not exist | 1.87±1.64 | 1.77±0.58 | 1.54±1.09 | 1.68±1.43 |
| | Moderate | 1.08±0.19 | 2.88±0.99 | 2.11±1.14 | 2.56±1.36 |
| | Severe | 1.89±1.41 | 2.78±0.96 | 2.50±1.34 | 2.75±1.41 |
| | P-value | 0.151 | 0.015* | 0.103 | 0.119 |
| Difficulty swallowing | Not exist | 1.88±1.49 | 2.51±0.83 | 2.29±1.32 | 2.63±1.49 |
| | Moderate | 1.28±0.50 | 2.96±1.29 | 2.70±1.21 | 2.58±1.29 |
| | Severe | 1.73±1.04 | 3.46±1.03 | 2.71±1.33 | 2.82±1.20 |
| | P-value | 0.444 | 0.000** | 0.322 | 0.836 |
| Weight loss | Not exist | 1.79±1.42 | 2.37±0.82 | 2.08±1.25 | 2.33±1.36 |
| | Moderate | 1.95±1.49 | 2.54±0.96 | 2.60±1.38 | 2.61±1.53 |
| | Severe | 1.77±1.26 | 3.29±0.93 | 2.73±1.31 | 3.13±1.34 |
| | P-value | 0.881 | 0.000** | 0.040* | 0.021* |
| Anxiety | Not exist | 1.52±1.16 | 2.35±0.82 | 1.95±1.19 | 2.25±1.38 |
| | Moderate | 1.47±0.98 | 2.71±1.01 | 2.51±1.46 | 2.54±1.37 |
| | Severe | 2.07±1.53 | 2.94±0.99 | 2.62±1.31 | 2.93±1.41 |
| | P-value | 0.084 | 0.012* | 0.046* | 0.062 |

(*Continued*)

**Table 11.** (Continued)

| Variables n = 120 | | Decline in basic taste | Discomfort (Taste disorder) | Phantogeusia and parageusia | General TAs |
|---|---|---|---|---|---|
| | | Mean ± SD | Mean ± SD | Mean ± SD | Mean ± SD |
| Excessive thirst | Not exist | 1.77±1.33 | 2.36±0.82 | 1.98±1.08 | 2.32±1.32 |
| | Moderate | 1.45±1.05 | 2.62±1.19 | 2.20±1.14 | 2.31±1.12 |
| | Severe | 1.90±1.45 | 3.07±0.97 | 2.81±1.43 | 3.03±1.47 |
| | P-value | 0.657 | 0.000** | 0.004* | 0.024* |
| Difficulty concentrating | Not exist | 1.65±1.28 | 2.55±0.96 | 2.01±1.15 | 2.26±1.33 |
| | Moderate | 1.75±1.42 | 2.72±0.82 | 2.66±1.54 | 2.69±1.47 |
| | Severe | 1.97±1.45 | 2.88±1.01 | 2.68±1.34 | 3.03±1.41 |
| | P-value | 0.478 | 0.238 | 0.025* | 0.020* |
| Lack of energy | Not exist | 1.38±0.96 | 2.21±1.04 | 1.74±1.01 | 1.65±0.96 |
| | Moderate | 1.74±1.46 | 2.04±0.55 | 2.38±1.56 | 2.46±1.73 |
| | Severe | 1.87±1.41 | 2.84±0.95 | 2.48±1.33 | 2.81±1.40 |
| | P-value | 0.479 | 0.015* | 0.164 | 0.019* |
| Confusion | Not exist | 1.84±1.45 | 2.60±0.96 | 2.23±1.35 | 2.53±1.46 |
| | Moderate | 1.53±0.97 | 2.81±0.73 | 2.37±0.84 | 2.63±0.96 |
| | Severe | 1.81±1.25 | 3.03±1.03 | 2.87±1.28 | 3.05±1.38 |
| | P-value | 0.813 | 0.125 | 0.083 | 0.243 |

Significant at *: p<0.05, **: p<0.001 according to one-way ANOVA test.

assessment method. Otherwise, our findings indicated a significant association between TAs and weight loss, as it is observed to be increased among individuals with TA.

Previous findings on the association between TA and malnutrition are limited and controversial. One study found that half of patients with TAs were malnourished or at risk of

**Table 12. Effect of comorbidities on taste alterations.**

| Variables n = 120 | | Decline in basic taste | Discomfort (Taste disorder) | Phantogeusia and parageusia | General TAs |
|---|---|---|---|---|---|
| | | Mean ± SD | Mean ±SD | Mean ± SD | Mean ± SD |
| CCI score | Mild (1,2) | 1.80±1.45 | 2.33±0.80 | 2.41±1.37 | 2.55±1.56 |
| | Moderate (3,4) | 1.90±1.40 | 3.24±1.13 | 2.66±1.45 | 2.95±1.43 |
| | Severe (≥5) | 1.78±1.29 | 2.85±0.90 | 2.23±1.19 | 2.63±1.25 |
| | P-value | 0.935 | 0.000** | 0.432 | 0.517 |
| Pulmonary disease | Yes | 3.00±1.87 | 2.46±0.96 | 2.40±1.94 | 2.70±1.78 |
| | No | 1.76±1.33 | 2.73±0.98 | 2.40±1.30 | 2.66±1.41 |
| | P-value | 0.049* | 0.547 | 1.000 | 0.958 |
| Diabetes mellitus | Yes | 1.52±0.97 | 3.18±1.05 | 2.31±1.27 | 2.75±1.10 |
| | No | 1.88±1.45 | 2.61±0.92 | 2.42±1.34 | 2.64±1.49 |
| | P-value | 0.250 | 0.009* | 0.741 | 0.750 |
| DM end organ | Yes | 2.33±1.78 | 2.52±1.11 | 2.00±1.17 | 3.08±1.27 |
| | No | 1.78±1.38 | 2.73±0.97 | 2.42±1.33 | 2.64±1.43 |
| | P-value | 0.346 | 0.612 | 0.612 | 0.464 |
| Hypertension | Yes | 2.09±1.52 | 3.22±1.14 | 2.64±1.43 | 3.03±1.41 |
| | No | 1.71±1.30 | 2.53±0.83 | 2.30±1.27 | 2.52±1.40 |
| | P-value | 0.177 | 0.000** | 0.221 | 0.78 |

Significant at *: p<0.05, **: p<0.001 according to one-way ANOVA/independent t-test.

malnutrition, and malnourished patients had more severe taste changes [16], and TAs have a serious effect on eating behaviors among cancer patients which might lead to nutritional deficiencies [7, 10]. In contrast, a recent study found no statistically significant link between malnutrition and weight loss and taste and smell alterations [8], and dietary habits were not directly related to taste changes during chemotherapy [31]. Malnutrition during chemotherapy is attributed to metabolic changes such as inflammation, increased catabolism, anabolic resistance, and antineoplastic side effects like anorexia, nausea, and vomiting [32]. However, according to our findings, taste alteration is a contributing factor in developing malnutrition, though not the only one. Nevertheless, further studies are necessary to obtain more comprehensive results. Weight loss and TAs showed contradictory findings as well. While some studies found that weight loss occurred among patients who experienced impaired taste [33, 34], others found no link between TAs and weight changes [35].

Findings from our study did not find any association between TAs and biochemical measurements (hemoglobin, albumin, total protein, and C-reactive protein). However, in our findings, elevated levels of C-reactive protein were attributed to a systematic inflammatory state associated with poor nutritional status in cancer patients [36]. In addition, a low level of hemoglobin is a manifestation of anemia, where its incidence increases during chemotherapy [37].

Regarding BMI, our findings are in line with those of earlier research [12, 13, 35, 38] which found no appreciable differences in outcomes between patients with and without dysgeusia in terms of BMI. However, evidence confirms that the relationship between BMI and taste perception is complex, and the ability to recognize taste decreased as BMI increased [39, 40]. It is worth noting that the majority of our study participants showed a high percentage of being overweight or obese. This could be related to several factors other than nutritional status, including changes in body composition related to increased fluid retention, which is a common side effect of chemotherapy [41]. In addition, it is possible that patients had pre-existing obesity or were overweight, as indicated by the non-significant difference between their body weight before and after CT in our study. Moreover, approximately fifty percent of the study participants are breast cancer patients, who usually tend to gain weight after breast cancer diagnosis due to decreased energy expenditure, hormonal imbalance, and depression, according to previous findings [42].

## Taste alteration risk factors

**Sociodemographic and lifestyle.** Chemotherapy-induced TA was affected by patients' sociodemographic characteristics. In this study, age was found to be associated with basic taste reduction, taste disorder, and general taste alteration. Basic taste reduction and general taste alteration mean scores were higher in patients aged between 36 and 55 years, while taste disorder score increased as age increased. Similar studies using CiTAS for self-reported taste impairment found that age did not affect TAs [12], while others found that age is associated with the basic taste reduction subscale [24], and TAs onset was associated with younger age [43]. Evidences revealed that older adults reported fewer TAs and late taste perception due to increased taste thresholds [2, 44]. Aging also reduces taste acuity, which diminishes after 60 due to aging-related physiology, reduced taste receptors, and some elderly medications [44].

Our study found that female participants had higher mean scores in phantogeusia and parageusia subscales exclusively, although a previous study found that taste disorder subscale mean was greater in women [24]. Arikan et al. find no link between gender and TAs subscales. However, female patients are more sensitive to TAs than male, although the reasons are not clearly known [12]. In addition, educational level was found to be associated with the taste discomfort

subscale, and patients with no school level presented the highest mean score. Our result was found to be contrary to recent studies findings [13, 16], which suggests that taste changes have been found to be more common in younger patients with higher education levels. Individuals with a high level of education are likely to be more sensitive to taste changes and to recognize them more rapidly [16]. Variations between our results and prior findings were assumed to be related to sample size, as more than 70% of participants are at school level, making them more susceptible to TAs.

Regarding lifestyle factors, TAs were significantly associated with sleeping problems for taste disorders, phantogeusia and parageusia, and general TAs subscales. Taste changes can have a psychological impact on a patient's lifestyle due to the loss of enjoyment during food intake and the pleasure of eating, as well as other CT side effects that cause anxiety and insomnia that disrupt their sleeping pattern. In a recent study on taste changes and functional status, patients with taste changes had more sleeping problems [3]. Physical activity, on the other hand was associated with phantogeusia, parageusia, and general TAs. Patients who had the least physical activity had higher mean scores. In line with our findings, previous studies found an association between TAs and being tired or fatigued [4, 10]. In addition, our findings revealed no association between smoking status and TAs. However, smoking has a negative impact on taste functions [24, 43], which is assumed to be due to having a higher threshold than nonsmokers and reduced taste sensitivity [45]. Therefore, smokers were shown to be less affected by chemotherapy-induced taste changes [2, 10].

**Clinical features (cancer-related data).** In our study, type of cancer was associated with phantogeusia and parageusia, whereas breast cancer patients had the highest mean score, followed by lymphoma, colorectal cancer, and lung cancer. Similar studies that used CiTAS found no link between type of cancer and TAs [12, 24]. In contrast, Ponticelli et al. discovered that TA is associated with cancer type [13].

In addition, our results showed that chemotherapy regimens did cause significant variations in phantogeusia and parageusia in patients receiving taxane-based chemotherapy as well as adriamycin and cyclophosphamide chemotherapy. In addition, taxane-based chemotherapy showed significant variations in the general taste alteration subscale. Taxane derivatives (paclitaxel and docetaxel) are mostly used for breast cancer treatment, which might explain the high prevalence of phantogeusia and parageusia among breast cancer patients. In previous studies [2, 13, 46, 47], taxane-based chemotherapy was associated with a higher rate of TAs. Still, no mechanism is documented to explain how taxane induces taste alterations [30]. Otherwise, cyclophosphamide has two mechanisms for taste disturbance: direct and indirect. It was shown that cyclophosphamide directly induces the destruction of the lingual epithelial cells, resulting in the death of sensory cells in taste buds, increasing the taste threshold and decreasing the ability of taste discrimination. Furthermore, when aged gustative cells die, cyclophosphamide indirectly prevents their replacement by suppressing the normal taste cell replacement process [30, 48].

The number of completed chemotherapy cycles results in significant variation in basic taste reduction as well as phantogeusia and parageusia. However, the mean score pattern is quite fluctuating. A previous study showed that TAs were associated with the number of cycles of chemotherapy in an increasing manner [11]. Another study reported that patients' ability to taste decreased directly after the first round of chemotherapy [14], and patients were found to have more severe taste impairments at the beginning of the treatment compared to the subsequent rounds [49].

**Comorbidities.** In the present study, we found that the number of comorbidities may increase the risk of TA, and patients with pulmonary disease, diabetes, and hypertension represent higher levels of TAs.

Chronic diseases are one of many factors that influence chemotherapy-induced TAs among patients who have moderate taste alterations [49]. A new study found that chronic obstructive pulmonary disease (COPD) and asthma patients are more susceptible to develop oral disorders, including TAs during inhalation therapy [50], which may explain our findings. On the other hand, through decades, patients with diabetes showed a high prevalence of TAs [51–53]. To the best of our knowledge, no previous studies have investigated the relationship between TAs and hypertension, but one study discovered that a high prevalence of hypertension was associated with impaired salt taste perception [54].

### Antineoplastic side effects and taste alteration relationship

In the present study, patients with nausea and vomiting showed higher levels of TAs than patients without nausea and vomiting, which is consistent with earlier findings [12, 55]. Furthermore, patients with dry mouth, sore mouth and throat, excessive thirst, and swallowing difficulties showed higher levels of TAs. Dry mouth, or xerostomia, was found to be strongly associated with TAs in previous studies, and the presence of taste impairments was found to be significantly high in patients who reported xerostomia [14, 56], sore mouth and throat [46], and swallowing difficulties [24, 56]. In our findings, excessive thirst is thought to be related to xerostomia, and more investigation is required.

Consistent with previous findings [13, 24, 33], our results revealed that participants who had increased or poor appetite, weight loss, dizziness, lack of energy showed a higher levels of TA. In addition, patients with disturbed sleep, anxiety, and difficulty concentrating showed higher levels of TAs. However, recent evidence suggests that taste changes are associated with neurophysiological adverse effects, including anxiety, sleep disturbances, and impairment in cognitive function [3].

### Coping strategies

This study examined coping mechanisms for chemotherapy-treated cancer patients who were suffering TAs. Our data showed that patients were infrequently given any kind of self-care or management instruction for taste impairment symptoms. Furthermore, most patients were surprised to learn that TAs are a chemotherapy-induced side effect. The most common and effective technique among the suggested strategies was to consume flavorful protein foods such as poultry, white beans, and eggs. Eating bland food was another technique encountered by a few individuals to manage with TA, who discovered that food with fewer spices and low fiber content was more suitable for them. Another group of participants prefers to boil their food since it is more acceptable to them than consuming oil-based or spicy-rich foods. Cold food was also thought to offer them a better feeling in the mouth than hot food, and it was assumed to reduce the queasy feeling caused by taste impairments. Brushing one's teeth before eating, on the other hand, was observed to be ineffective by those who tried it. While half of the patients who tried to suck on hard candies and gum found it to be a beneficial strategy, otherwise, they indicated that as soon as they stopped doing so, their uncomfortable mouth sensation returned. It was also noted that patients rarely tended to have more frequent and smaller meals and often stuck to traditional cooking methods rather than trying out new ones. As an additional suggestion, patients stated that they prefer sour flavors over other flavors and consume lemon more frequently. Previous studies evaluating self-management strategies are limited. In a study that used the same data sheet we used in our study, it was discovered that coping strategies varied depending on the type of taste impairments patients complained of, and the most common coping strategies were eating blander food, eating frequent and smaller meals, oral care before eating, and avoiding foods with a strong smell and taste. Patients

recommend trying new flavors and avoiding certain foods, particularly hot and oily foods and those with tomato sauces [57]. In addition, patients avoided certain foods, rinsed their mouths frequently, ate cold foods, and avoided the sight and smell of some foods [58]. Furthermore, adding lemon, orange, or mint to drinking water, exploring new flavors, and practicing good oral hygiene were excellent self-care behaviors for individuals with TAs [59].

## Limitations

This study has some limitation like being subjective and self-reported study that relies on the patient's own statements which make them more susceptible to recall and selection bias due to the nature of cross-sectional studies. Another limitation is the small sample size and the lack of information regarding dietary habits obtained by diet recall, which was challenging for patients to get, as well as body circumferences due to the patients' physical status. Furthermore, participants in this experiment were given different chemotherapy regimens, with some receiving monotherapy and others receiving combination chemotherapy, making it difficult to distinguish the precise effect of individual chemotherapy on taste while ignoring the effect of combination chemotherapy.

## Conclusions

In this study, it was found that a significant proportion (98.3%) of cancer patients experienced taste changes during chemotherapy. Discomfort was an important determinant of the risk of malnutrition, and it showed the highest prevalence among participants, followed by general TA, phantogeusia and parageusia, and basic taste reduction. TAs are associated with the severity of a variety of antineoplastic side effects (nausea, vomiting, dry mouth, sour mouth, and throat; difficulty swallowing; excessive thirst; lack of energy; increased and poor appetite; weight loss; dizziness; distrusted sleep; anxiety; difficulty concentrating). Patients with pulmonary disease, diabetes, and hypertension were at greater risk of developing taste alterations. In light of these results, more attention is recommended at the educational, clinical, and research levels to present a better quality of life for cancer patients during their course of treatment and minimize the intensity of the chemotherapy-induced taste alterations. Furthermore, it could be beneficial for patients to actively engage in self-care by attentively monitoring their dietary habits and exploring new recipes. This proactive approach will allow them to identify food preferences that they are more likely to eat after chemotherapy, thereby mitigating the adverse effects of weight loss, decreased appetite, and risk of malnutrition.

## Acknowledgments

We would like to thank each patient who participates in this work.

## Author Contributions

**Methodology:** Manal Badrasawi.

**Writing – original draft:** Fatima Masoud Al-Amouri, Manal Badrasawi.

**Writing – review & editing:** Fatima Masoud Al-Amouri, Manal Badrasawi.

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
