## [Decision Letter · Decision Letter 0]

26 Dec 2023

PONE-D-23-31394Taste alteration and its relationship with nutritional status among cancer patients receiving chemotherapy, cross-sectional study.PLOS ONE

Dear Dr. Badrasawi,

Thank you for submitting your manuscript to PLOS ONE. After careful consideration, we feel that it has merit but does not fully meet PLOS ONE’s publication criteria as it currently stands. Therefore, we invite you to submit a revised version of the manuscript that addresses the points raised during the review process.

We look forward to receiving your revised manuscript.

Kind regards,

Naim Akhtar Khan, PhD, DSc

Academic Editor

PLOS ONE

4. Please upload a copy of Supporting Information Figures (S1 Fig.1 and S2 Fig.2) which you refer to in your text on page 33.

Additional Editor Comments:

The MS should be revised as per comments of the reviewer.

Reviewers' comments:

Reviewer's Responses to Questions

**Comments to the Author**

1. Is the manuscript technically sound, and do the data support the conclusions?

Reviewer #1: Yes

Reviewer #2: Yes

2. Has the statistical analysis been performed appropriately and rigorously? 

Reviewer #1: Yes

Reviewer #2: Yes

3. Have the authors made all data underlying the findings in their manuscript fully available?

Reviewer #1: Yes

Reviewer #2: Yes

4. Is the manuscript presented in an intelligible fashion and written in standard English?

Reviewer #1: Yes

Reviewer #2: Yes

5. Review Comments to the Author

Reviewer #1: The article titled “Taste alteration and its relationship with nutritional status among cancer patients receiving chemotherapy: a cross-sectional study” aimed to determine the prevalence of taste alterations during chemotherapy and their association with nutritional status and malnutrition. The study is interesting; the Authors determined different aspects relating to the problem using different questionnaires, however the paper is difficult to follow. In particular, the methods are not detailed enough, and this creates problems for the readers. There are some points that need to be carefully elucidated before publication.

Introduction

In lines 49-50: The author’s state: “and patients exhibiting increased sensitivity to it (6).” In reference 6, authors declare that sweet thresholds significantly declined in patients. Please correct the statement.

Methods

The authors evaluate different parameters by using specific scales or tools: TAs were evaluated using the chemotherapy-induced taste alteration scale (CiTAS), the malnutrition universal screening tool (MUST) was used for nutritional screening, the antineoplastic side effects scale (ASES) was used for subjective assessment of chemotherapy side effects, and the Charlson comorbidity index (CCI) was used for comorbidity assessment. It is not clear why for each scale or tools the author’s used the Cronbach’s alpha test to evaluate the reliability. Please explain.

Table 2:

In the methods Authors indicate that the Charlson comorbidities index includes 17 comorbidities, why Authors include 20 comorbid conditions in the table? Please explain or correct.

What the “assigned weighing” means?

Table 3 includes Patient’s biochemical levels presented in n (%). The number of subjects is different for each parameter. Why have the Authors choice to include these parameters (albumin, hemoglobin, total protein and c-reactive protein)? and why Authors report these data only in few patients? Have these parameters an association with specific cancer? Please explain the reason for your choice. Did the Authors consider that these parameters are involved in Taste alteration or malnutrition? Please explain. If there is no specific reason, or any association with the study I would suggest removing these data.

The name Table 4 is used twice (line 176 and line 184) please correct it.

Regarding Table 4 Chemotherapy side effects. I would suggest indicating the side effects in order of greatest percentage of frequency.

Reviewer #2: The MS "Taste alteration and its relationship with nutritional status among cancer patients

receiving chemotherapy, cross-sectional study." is well written and well organized with an in-depth approach and including several parameters that can modify taste in cancers

6. PLOS authors have the option to publish the peer review history of their article (what does this mean?). If published, this will include your full peer review and any attached files.

Reviewer #1: No

Reviewer #2: No

---

## [Author Response · Author response to Decision Letter 0]

23 Jan 2024

We kindly thank you for your time and effort invested during the review of our manuscript. We appreciate the comments and suggestions on the submitted version of our contribution. We did all the corrections suggested by the reviewers attached to this letter in point-to-point form, all the corrections have been approved by all authors. The changes in the manuscript were mentioned in a table also they were highlighted in yellow in the manuscript for easier tracing. 

We hope that the answer letter and the improved version of the paper will clarify open issues and we are looking forward to additional feedback on our revised proposal.

Kind regards,

The authors team.

---

## [Decision Letter · Decision Letter 1]

4 Apr 2024

PONE-D-23-31394R1Taste alteration and its relationship with nutritional status among cancer patients receiving chemotherapy, cross-sectional study.PLOS ONE

Dear Dr. Badrasawi,

Thank you for submitting your manuscript to PLOS ONE. After careful consideration, we feel that it has merit but does not fully meet PLOS ONE’s publication criteria as it currently stands. Therefore, we invite you to submit a revised version of the manuscript that addresses the points raised during the review process.

We look forward to receiving your revised manuscript.

Kind regards,

Naim Akhtar Khan, PhD, DSc

Academic Editor

PLOS ONE

Journal Requirements:

Additional Editor Comments:

The MS must be revised with minor corrections as report.

Reviewers' comments:

Reviewer's Responses to Questions

**Comments to the Author**

1. If the authors have adequately addressed your comments raised in a previous round of review and you feel that this manuscript is now acceptable for publication, you may indicate that here to bypass the “Comments to the Author” section, enter your conflict of interest statement in the “Confidential to Editor” section, and submit your "Accept" recommendation.

Reviewer #3: All comments have been addressed

2. Is the manuscript technically sound, and do the data support the conclusions?

Reviewer #3: Yes

3. Has the statistical analysis been performed appropriately and rigorously? 

Reviewer #3: Yes

4. Have the authors made all data underlying the findings in their manuscript fully available?

Reviewer #3: Yes

5. Is the manuscript presented in an intelligible fashion and written in standard English?

Reviewer #3: Yes

6. Review Comments to the Author

Reviewer #3: Reviewer_Pr. EA. KOCEIR

Head of the Natural and Life Sciences Domain - Head of Nutrition and Pathologies Post Graduate School

Head of Nutrition and Human Dietetics Master - Head of Bioenergetics and Intermediary Metabolism team

Biology and Organisms Physiology laboratory - Faculty of Biological Sciences - University of Sciences and Technology "Houari Boumédiene" (USTHB) - BP 32, Elalia, 16111, Bab Ezzouar, Algiers, ALGERIA - Tel/Fax : [+ 213] (0) 21 24 72 17 - Cell phone : [+ 213] (0)6 66 74 27 70

Alternative email address : ekoceir@usthb.dz; ea.koceir@outlook.fr

Google Scholar : https://scholar.google.fr/citations? - Sciprofiles : https://sciprofiles.com/profile/455388 - ORCID: 0000-0003-1345-2535

Peer Review for “Taste alteration and its relationship with nutritional status among cancer patients receiving chemotherapy, cross-sectional study"

PONE-D-23-31394R1

PLOS ONE

Dear Dr. Koceir,

Thank you for agreeing to submit a review on PLOS ONE manuscript "Taste alteration and its relationship with nutritional status among cancer patients receiving chemotherapy, cross-sectional study.." As a reminder, your review is due by ven. 15 mars 2024 (UTC+1) can be submitted at https://www.editorialmanager.com/pone/.

Dear Professor Naim Akhtar Khan, Academic Editor

I'm sorry I didn't answer right away. I was busy with a national examination period to enter postgraduate school, which required me to orally examine candidates.

General view

Authors have investigated the relationship between altered taste perceptions and the prevalence of denutrition in cancer patients receiving chemotherapy. Overall, this is an original clinical study, both in the scientific theme related to the physiology of taste dysfunction, and in the complex cancer pathology. The study was fine conducted both by methodology approach and statistically cohort. The MS is very well written with good academic English. Regarding the review, I have read that comments have been raised by other reviewers and there is no need to ask the same questions again. However, some comments relate specifically to the nutritional status.

Comments

1. In lines 184-185: Thanks to set the unit of the value body weight (79.43±17.34! and 77.41±17.23 !). It seems that the difference is not significant before and after chemotherapy. It should be remembered that the BMI is an index of corpulence and not of malnutrition. If we consider the BMI, your patients do not present malnutrition (0.83% were underweight in line 184). Moreover 70% are obese-overweight. Why?

2. This same observation (non-significant) is noted for the malnutrition serum parameters (table 3) such as albumin (10.9%) and total proteins (14.28%). Haemoglobin and C-reactive protein is not linked to malnutrition, but to anaemia and inflammation, respectively. Thank you for giving arguments

3. I am not opposed to the MUST test, but it is admitted that it is the Nutritional Risk Index (NRI) test or Buzby index which is the most practiced. In your study we do not know the degree of malnutrition: severe or moderate. Thank you for your comment.

4. If we consider that cancer patients undergoing chemotherapy present malnutrition. Is this due to anorexia (linked to cytokine storm) or to altered taste perception!!

7. PLOS authors have the option to publish the peer review history of their article (what does this mean?). If published, this will include your full peer review and any attached files.

Reviewer #3: No

---

## [Author Response · Author response to Decision Letter 1]

10 Apr 2024

We kindly thank you for your time and effort invested during the review of our manuscript. We appreciate the comments and suggestions on the submitted version of our contribution. We did all the corrections suggested by the reviewers attached to this letter in point-to-point form, all the corrections have been approved by all authors. The changes in the manuscript were mentioned in the below table also they were highlighted in yellow in the manuscript for easier tracing. 

We hope that the answer letter and the improved version of the paper will clarify open issues and we are looking forward to additional feedback on our revised proposal.

Kind regards,

The authors team.

---

## [Editor Report · Decision Letter 2]

17 Apr 2024

Taste alteration and its relationship with nutritional status among cancer patients receiving chemotherapy, cross-sectional study.

PONE-D-23-31394R2

Dear Dr. Badrasawi,

We’re pleased to inform you that your manuscript has been judged scientifically suitable for publication and will be formally accepted for publication once it meets all outstanding technical requirements.

Kind regards,

Naim Akhtar Khan, PhD, DSc

Academic Editor

PLOS ONE

Additional Editor Comments (optional):

Please jur verify the style of the journal, if required.
---

## [Editor Report · Acceptance letter]

25 Apr 2024

PONE-D-23-31394R2 

PLOS ONE

Dear Dr. Badrasawi, 

I'm pleased to inform you that your manuscript has been deemed suitable for publication in PLOS ONE. Congratulations! Your manuscript is now being handed over to our production team.

Kind regards, 

on behalf of

Professor Naim Akhtar Khan 

Academic Editor

PLOS ONE